# Prevalence of glaucoma in Africa: A systematic review and Bayesian meta-analysis

Randy Asiamah[1,2], Samuel Kyei[1,2]*, Gideon Owusu[1], Patrick Evans Agyiri[1]

**1** School of Optometry and Vision Science, College of Health and Allied Sciences, University of Cape Coast, Cape Coast, Ghana, **2** Biomedical and Clinical Research Centre, College of Health and Allied Sciences, University of Cape Coast, Cape Coast, Ghana

* skyei@ucc.edu.gh

## Abstract

### Purpose

This study sought to establish the prevalence of glaucoma and the associated factors in Africa.

### Methods

A systematic review of studies reporting the prevalence of glaucoma was performed using three electronic databases: PubMed, Scopus, and Web of Science. Data were extracted and study-specific estimates of the prevalence of glaucoma were combined using meta-analysis to obtain pooled proportions.

### Results

A total of 9 studies that evaluated the prevalence of glaucoma in 29,606 individuals, comprising 14,487 males and 15,119 females were included in this study. The prevalence of glaucoma (unclassified glaucoma) in Africa was found to be 5.59% (95% Credible Interval (CrI) 4.32% to 7.74%). The prevalence of Primary Open-Angle Glaucoma (POAG), Primary Angle-Closure Glaucoma (PACG), and secondary glaucoma (SG) in Africa are 5.07% (95% (CrI) 3.51% to 8.52%), 0.98% (95% CrI 0.29% to 5.38%), and 2.19% (95% CrI 0.64% to 10.00%), respectively. The prevalence of glaucoma is highest in Southern Africa (6.47%, 95% CrI 3.10% to 12.10%) and lowest in East Africa (4.80%, 95% CrI 2.37% to 9.27%). The prevalence of POAG is highest in West Africa (6.48% 95% CrI 5.23% to 9.89%) and lowest in East Africa (3.23% 95% CrI 2.21% to 5.07%).

### Conclusion

The prevalence rates of glaucoma and POAG are high, with geographical regional variations worthy of note. Continued efforts are necessary to implement

**Data availability statement:** All relevant data are within the manuscript and its Supporting Information files.

**Funding:** The author(s) received no specific funding for this work.

**Competing interests:** The authors have declared that no competing interests exist.

population-based screening and public health education initiatives to foster early diagnosis and management.

## Introduction

Glaucoma is a group of diseases characterized by a progressive loss of retinal ganglion cells (RGCs). This loss results in unique changes to the tissues of the neuro-retinal rim of the optic nerve head (ONH), leading to visual field loss. Visual field loss can occur with or without an increase in intraocular pressure [1–4]. The main subtypes of glaucoma include primary glaucoma, i.e., Primary Open-Angle Glaucoma (POAG), Normal Tension Glaucoma (NTG), Primary Angle Closure Glaucoma (PACG) and secondary glaucoma (SG) [5–14]. Other forms include congenital and developmental glaucoma, which manifest in newborns and children [15].

Glaucoma has gained considerable notoriety as the leading cause of irreversible blindness and the second major cause of blindness after cataracts on a global scale [16–18]. A significant number of individuals who hitherto had good vision have had to succumb to the subtle yet damaging effects of glaucoma. It has hence gained the reputation of being the "silent thief of sight" [19]. It is usually painless and presents with no symptoms except at advanced stages.

Numerous risk factors which include advance age, a positive family history, thinner corneas, black ethnicity or African descent, systemic hypertension, diabetes, and myopia exist [20,21]. The distinctive narrative associated with glaucoma in different regions of the world necessitates the synthesis of available evidence to fully appreciate the prevalence and associated factors of this disease. For instance, it has been reported that in individuals of African descent, the disease is of early onset, severe, and more aggressive in that blindness is 6–8 times more likely compared to other races [22–24]. This evidence demonstrates that Africa is significantly impacted by glaucoma, yet there is a paucity of detailed information on the epidemiological nuances of the disease.

This study sought to review and analytically synthesize available data to establish the prevalence of glaucoma and the associated factors in Africa. This holds great potential in providing the needed information which may serve as a component contribution to the realization of sustainable development goal three (SDG 3) aimed at ensuring healthy lives and promoting well-being for all at all ages [25]. This will help inform efforts and strategies targeted at achieving the Universal Eye Health Coverage and to reduce untimely deaths resulting from blindness from glaucoma [26].

## Methods

This systematic review and meta-analysis has been registered on PROSPERO, with registration number CRD 42024511983, and was conducted in accordance with the Preferred Reporting Items for Systematic reviews and Meta-Analyses (PRISMA) statement guidelines [27]. The title provided in the registered protocol for this systematic review and meta-analysis was amended to align with the extant evidence.

## Eligibility criteria

Population-based cross-sectional or cohort studies published since the journals' inception till 25th January 2025 (date in which database searches were conducted) that reported the prevalence of clinically diagnosed glaucoma among African populations were included in this meta-analysis. Participants in the eligible studies involved persons of all age groups. The condition studied was glaucoma (and its subtypes). Studies were included if they met the following inclusion criteria: optic disc evaluation by slit-lamp biomicroscopy or fundus photography, visual field testing, anterior chamber angle/depth evaluation by slit-lamp biomicroscopy or gonioscopy. Case definitions of glaucoma and its subtypes were based on structural or functional evidence of glaucomatous optic neuropathy as assessed by optic disc evaluation or visual field testing, respectively.

Studies in which the comparators were individuals who were unaffected by Glaucoma were included. The main outcome considered was the prevalence of glaucoma and its subtypes.

Non-English language studies were included and were translated into English using Google Translate. Primary studies that were either not fully published in peer-reviewed journals, not conducted in individuals of African descent living on the African continent, or with self-reported diagnosis of glaucoma were excluded. Studies with unavailable full-text and those that repeated data from the author's previous publications were also excluded. Narrative reviews, discussion papers, non-research letters or editorials, case series, case reports, and animal studies were excluded. Primary studies with significant amounts of missing data were excluded from the analysis.

## Information sources

Information was retrieved from major bibliographic databases (PubMed, Scopus, and Web of Science). The bibliographic searches retrieved studies published since the journals' inception. A detailed outline of the search strategies used for each database is available in a supporting file (S1 Table). The reference lists of each of the selected articles were screened to identify other potentially relevant studies.

## Selection process

Relevant studies published since journal inception that were retrieved from the database searches were screened first by title and abstract, and then by full text, by two independent reviewers (GO and PEA). Disagreements between the pair were resolved by consensus or consultation with the project PI (SK). The Preferred Reporting Items for Systematic Reviews and Meta-Analysis (PRISMA) flow diagram was employed to report the screening process (Fig 1).

## Data collection process

Two reviewers (GO and PEA) extracted data from each eligible study using a standardized data extraction sheet and subsequently cross-checked the results. Data extracted included first author's name, year of study, country, participant characteristics including age and sex, number of subjects, period of data collection, disease subtype, and prevalence. The degree to which the glaucoma diagnosis in the included studies conformed to the International Society for Geographical and Epidemiological Ophthalmology (ISGEO) criteria [28] was documented. Furthermore, the present study documented whether visual field assessments were conducted for the entire sample, a subset of high-risk participants, or a proportion of random participants of the included studies. Disagreements regarding the extracted data were resolved through discussion with a third reviewer (SK).

## Data synthesis and analysis

Study-specific estimates of crude prevalence, calculated from the number of individuals with glaucoma (events) and the total population (sample) from each study, were combined to estimate pooled effect sizes ($\mu$) for the prevalence of

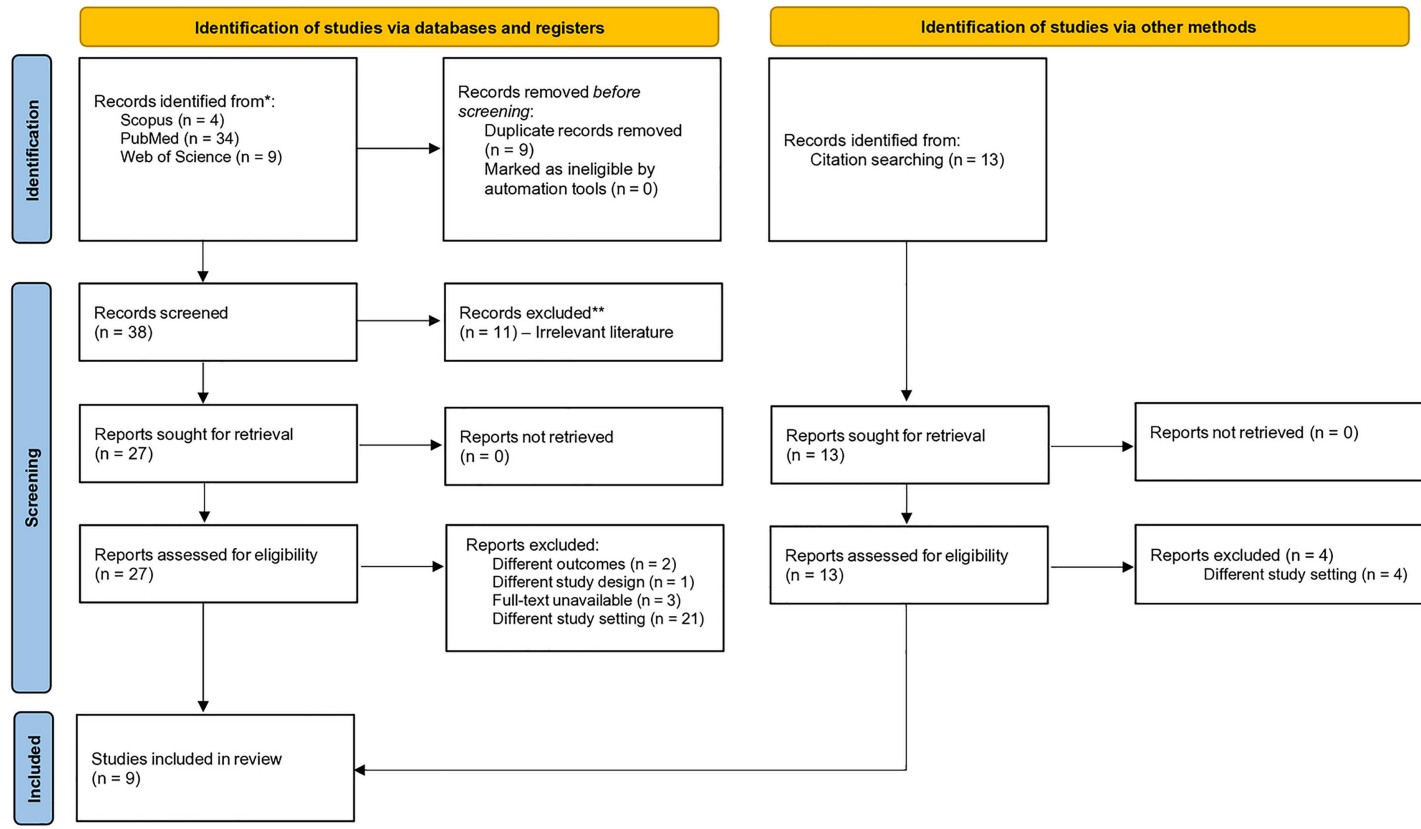

**PRISMA 2020 flow diagram for new systematic reviews which included searches of databases, registers and other sources**

Source: Page MJ, et al. BMJ 2021;372:n71. doi: 10.1136/bmj.n71.

**Fig 1. PRISMA flowchart.**

glaucoma and its subtypes using the Bayesian random-effects meta-analysis approach. Bayesian meta-analysis was chosen over frequentist methods because it facilitates the incorporation of prior knowledge, which is particularly valuable in situations where individual studies are limited in number or exhibit significant heterogeneity [29]. This framework yields credible intervals that provide direct probabilistic interpretations of uncertainty, offering more intuitive insights compared to traditional confidence intervals [29]. Moreover, the flexibility of Bayesian hierarchical models efficiently captures between-study variability, ensuring that estimates remain robust under various model configurations and prior assumptions [29]. Importantly, the Bayesian approach also provided a direct comparison to a similar study conducted by Tham et al. [30], which employed Bayesian meta-analysis methods to estimate glaucoma prevalence, allowing for meaningful alignment and comparison of findings across studies.

Since an informative prior was deemed necessary to precisely estimate heterogeneity [31]. the half-normal distribution with scale, 0.5, was applied. Informative priors, such as the half-normal distribution with a scale of 0.5, were chosen to precisely estimate heterogeneity by leveraging previous research recommendations. [32,33] This prior reflects the belief that heterogeneity is expected to be non-negative and relatively small, thereby stabilizing the estimates in the presence of limited data [32,33]. For log-odds ratio endpoints (odds ratios expressed on a logarithmic scale), informative priors are similarly advised, and a sensitivity analysis was performed using alternative priors—namely, a half-normal distribution with a scale of 1.0 and a half-Cauchy distribution with a scale of 0.5—to test the robustness of the findings [32,33]. The

credible intervals (CrI), defined by the 2.5th to 97.5th percentiles of the posterior distributions, transparently capture the uncertainty inherent in the model while reflecting the balance between prior knowledge and observed data [34,35]. CrI is a range derived from Bayesian statistics that tells us where we believe the true value of a parameter likely lies, based on the data and prior beliefs [34,35].

A Bayesian meta-regression analysis was conducted to investigate potential factors underpinning variance and heterogeneity in the main meta-analysis by evaluating joint and marginal posterior probability distributions for the included covariates. The most consistently reported parameters, geographic location and year of publication were chosen as covariates for the meta-regression.

The equation used for the main Bayesian analysis model is computationally expressed as follows:

$$\text{cases}|\text{trials}(n) \sim 1 + (1|\text{Author})$$

The Bayesian regression model is as follows:

$$\text{cases}|\text{trials}(n) \sim 1 + \text{Location} + \text{Year} + (1|\text{Author})$$

The term, "cases" was used to represent the number of individuals diagnosed with glaucoma) or its subtypes), "trials(n)" represents the total sample size, "Location" was used as a categorical predictor for study geographical region in Africa, "Year" was used as a continuous predictor representing year of study, and (1|Author) was used to denote a random intercept to account for between-study variability.

Statistical analysis was performed using R (version 4.4.3), with the *brms* package used for the Bayesian meta-analysis, Bayesian meta-regression, and for sensitivity analysis. All Bayesian model analyses were set with 4000 iterations. Posterior distributions were obtained through Markov Chain Monte Carlo (MCMC) sampling.

Publication bias was not assessed, as the number of studies corresponding to each glaucoma subtype was less than 10 [36].

### Study risk of bias assessment

Two reviewers (RA and SK) employed the use of the Joanna Briggs Institute Prevalence Critical Appraisal Tool (JBI-PCAT) in evaluating the quality of the included studies [37]. The JBI-PCAT has been proposed as the most appropriate tool for assessing the methodological quality of prevalence studies. It contains nine domains, which are answered with Yes, No, Unclear, and Not applicable. According to the number of "Yes", the studies were classified into three levels – high quality ≥ 6; moderate quality 4–5; low quality ≤ 3.

## Results

Database searches yielded 47 potentially relevant entries with 13 studies obtained from citation searching. After removing duplicated publications, 51 articles remained for title and abstract screening. Out of 51 titles and abstracts screened, 40 were sought for full-text retrieval, and 31 were excluded for various reasons, including different study outcomes (2 studies), different study design (one study), different study settings (hospital-based studies, 25 studies) and unavailable full text (three studies). Finally, nine studies met the inclusion criteria and were included in the systematic review and meta-analysis. A list of all studies identified in the literature search including excluded studies and reasons for their exclusion is shown in S2 Table.

### Study characteristics

The articles included in this study were published between 2000 and 2019 (Table 1). The total sample size was 29,606, comprising 14,487 males and 15,119 females. Three studies were conducted in Nigeria, two studies in Ghana, two studies

in South Africa, and one study each from Tanzania and Kenya. A total of nine studies identified 1,604 cases (out of a sample size of 29,606) of glaucoma (unclassified glaucoma), six studies identified 667 cases of POAG (out of a sample size of 13,290), four studies identified 31 cases of PACG (out of a sample size of 10,666), and four studies identified 45 cases of SG (out of a sample size of 10,666). A total of six studies reported at least one subtype of glaucoma in addition to the glaucoma (unclassified glaucoma) category. Table 1 summarizes characteristics of the nine included studies [38–46]. Data extracted from the nine primary sources for the present study are provided in S3 Table.

### Risk of bias assessment

All nine studies included in this review were found to be of low risk of bias, with one study (by Ekwerekwu et al. [40]) satisfying eight out of the nine JBI-PCAT items as it failed to describe study subjects and setting in detail. The robust methodological quality across the studies reinforces confidence in the reliability of their findings (S1 Fig).

### Prevalence of Glaucoma

Table 2 provides a summary of the estimates of heterogeneity (τ) and true effect (µ) for the main meta-analysis conducted using the half-normal informative prior with a scale of 0.5. The results of the sensitivity analysis are also presented in Table 2.

The prevalence of glaucoma (unclassified glaucoma) in Africa was found to be 5.59% (95% CrI 4.32% to 7.74%) (Fig 2). The prevalence of POAG, PACG, and SG in Africa are 5.07% (95% CrI 3.51% to 8.52%) (Fig 3), 0.98% (95% CrI 0.29% to 5.38%) (Fig 4), and 2.19% (95% CrI 0.64% to 10.00%) (Fig 5), respectively.

**Table 1. Characteristics of included studies.**

| Author | Country | Geographical location | Male | Female | Age range (years) | ISGEO | VF/IOP |
|---|---|---|---|---|---|---|---|
| Buhrmann 2000 [38] | Tanzania | East Africa | 1437 | 1810 | 40+ | 3 | 0 |
| Rotchford 2002 [39] | South Africa | Southern Africa | 725 | 280 | 40+ | 1 | 0 |
| Ekwerekwu 2002 [40] | Nigeria | West Africa | 206 | 458 | 30+ | 3 | 2 |
| Rotchford 2003 [41] | South Africa | Southern Africa | 280 | 559 | 40-97 | 1 | 0 |
| Ntim-Amponsah 2004 [42] | Ghana | West Africa | 892 | 893 | 30+ | 3 | 2 |
| Ashaye 2013 [43] | Nigeria | West Africa | 389 | 422 | 40+ | 2 | 3 |
| Budenz 2013 [44] | Ghana | West Africa | 2224 | 3379 | 40+ | 2 | 3 |
| Kyari 2015 [45] | Nigeria | West Africa | 7345 | 6246 | 40+ | 1 | 1 |
| Bastawrous 2019 [46] | Kenya | East Africa | 989 | 1072 | 50+ | 1 | 0 |

**ISGEO: International Society for Geographical and Epidemiological Ophthalmology Classification**

1: "Follows ISGEO & VF on all" (Study design follows the ISGEO criteria and visual field assessment was performed on all participants).

2: "Follows ISGEO & VF on subset" (Study design follows the ISGEO criteria and visual field assessment was performed on a subset of participants limited to a proportion of subjects or only high-risk subjects).

3: "Does not follow ISGEO" (A more conventional method of determining glaucoma using a combination of optic disc features & visual field defects.

**VF/IOP: Visual Field/Intraocular Pressure Classification**

0: "VF on all" (Diagnosis of glaucoma included visual field assessment on all participants and intra ocular pressure was not used as a defining criterion of glaucoma).

1: "VF on all & IOP criterion" (Diagnosis of glaucoma included visual field assessment on all participants and intra ocular pressure was used as a defining criterion of glaucoma).

2: "IOP criterion & VF on subset" (Diagnosis of glaucoma did not include visual field assessment on all participants and intra ocular pressure was used as a defining criterion of glaucoma).

3: "Other" (Diagnosis of glaucoma did not include visual field assessment on all participants nor did glaucoma case definition rely on intra ocular pressure criteria).

**Table 2. Posterior summary of estimates for the main meta-analysis and for all sensitivity analyses.**

| Glaucoma Type | Prior | Log-Odds Estimate (95% CrI) | Between-Study Variability (95% CrI) |
|---|---|---|---|
| Glaucoma (unclassified glaucoma) | Half-Normal (0.5) | −2.83 (−3.10, −2.48) | 0.41 (0.22, 0.73) |
| | Half-Normal (1.0) | −2.82 (−3.13, −2.47) | 0.44 (0.23, 0.79) |
| | Half-Cauchy (0.5) | −2.82 (−3.13, −2.46) | 0.43 (0.23, 0.78) |
| POAG | Half-Normal (0.5) | −2.93 (−3.31, −2.37) | 0.45 (0.22, 0.88) |
| | Half-Normal (1.0) | −2.88 (−3.35, −2.12) | 0.56 (0.24, 1.31) |
| | Half-Cauchy (0.5) | −2.91 (−3.31, −2.30) | 0.49 (0.22, 1.21) |
| PACG | Half-Normal (0.5) | −4.61 (−5.85, −2.87) | 1.00 (0.29, 1.85) |
| | Half-Normal (1.0) | −2.55 (−4.70, −0.40) | 2.40 (1.09, 3.67) |
| | Half-Cauchy (0.5) | −0.95 (−3.16, 1.14) | 5.70 (2.28, 12.94) |
| SG | Half-Normal (0.5) | −3.80 (−5.05, −2.20) | 1.24 (0.69, 1.92) |
| | Half-Normal (1.0) | −2.53 (−4.42, −0.60) | 2.19 (1.05, 3.51) |
| | Half-Cauchy (0.5) | −1.18 (−3.56, 1.03) | 4.66 (1.64, 10.70) |

POAG – Primary Open-Angle Glaucoma; PACG – Primary Angle-Closure Glaucoma; SG – Secondary Glaucoma.

**Credible Interval (CrI)** is a range derived from Bayesian statistics that tells us where we believe the true value of a parameter likely lies, based on the data and prior beliefs.

**Half-Normal (0.5):** uses a half-normal distribution with a scale of 0.5. A half-normal distribution is simply a normal distribution restricted to nonnegative values, giving higher weight to values near zero, which is useful when assuming that the parameter (e.g., between-study variability) is likely to be small but must be nonnegative.

**Half-Normal (1.0):** assigns a half-normal distribution but with a scale of 1.0. The larger scale means that it is less informative—allowing for greater variability—while still ensuring that only nonnegative values are considered.

**Half-Cauchy (0.5):** uses a half-Cauchy distribution with a scale of 0.5. The half-Cauchy distribution is also restricted to nonnegative values and is known for its heavier tails compared to the half-normal. This makes it a more robust choice when there is a possibility of larger parameter values, as it accommodates greater uncertainty.

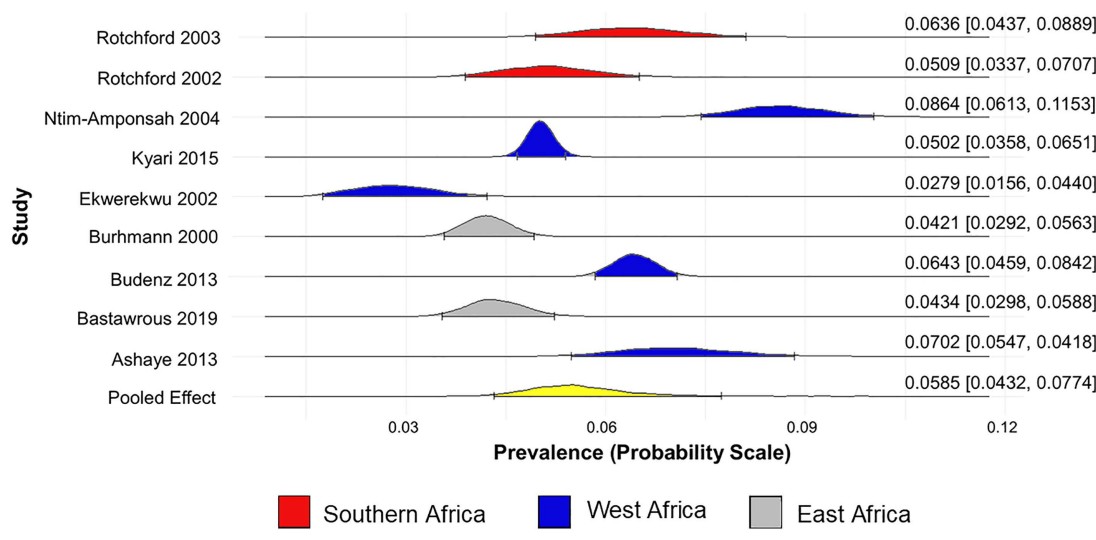

**Fig 2. Forest plot of the prevalence of glaucoma (unclassified glaucoma) in Africa.**

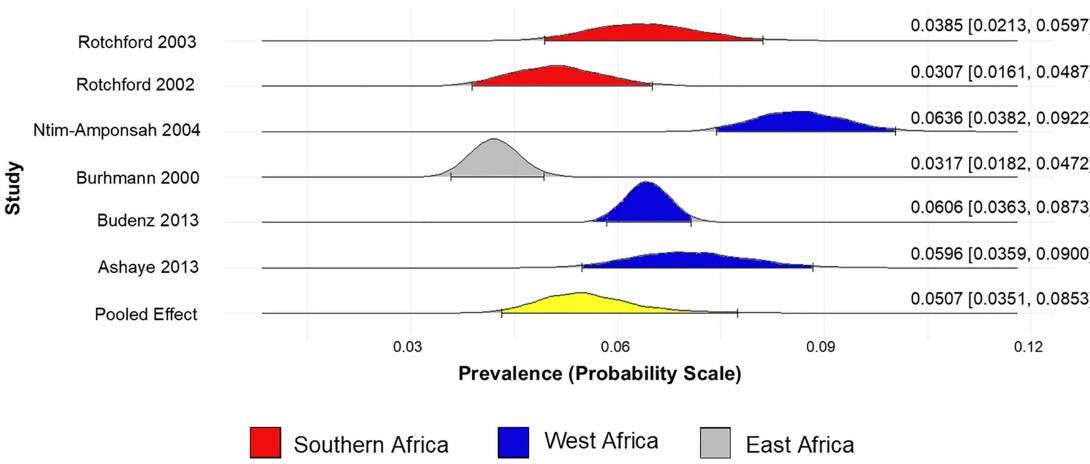

**Fig 3. Forest plot of the prevalence of primary open-angle glaucoma in Africa.**

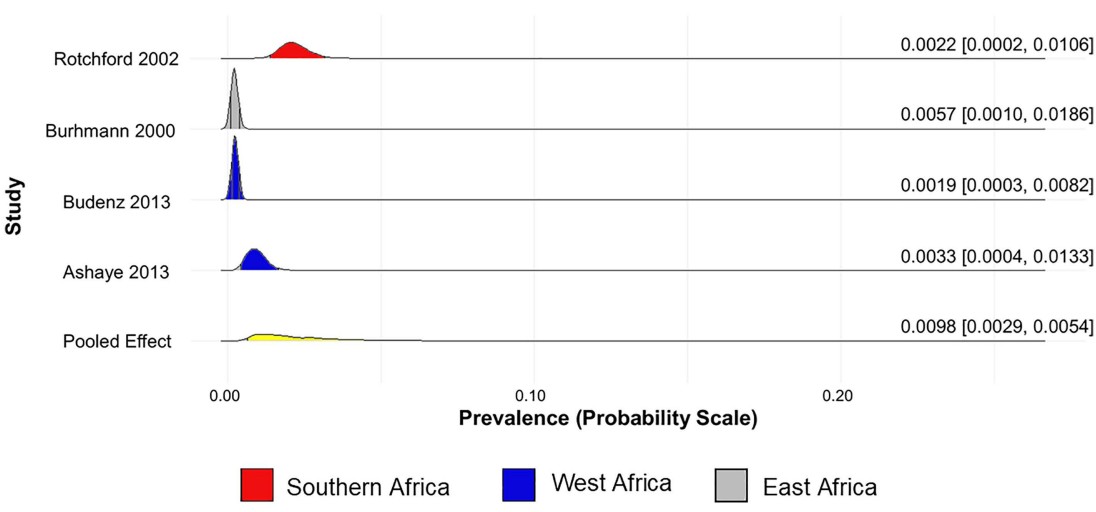

**Fig 4. Forest plot of the prevalence of primary angle-closure glaucoma in Africa.**

The prevalence of glaucoma is highest in Southern Africa (6.47%, 95% CrI 3.10% to 12.10%) and lowest in East Africa (4.80%, 95% CrI 2.37% to 9.27%). The prevalence of POAG is highest in West Africa (6.48% 95% CrI 5.23% to 9.89%) and lowest in East Africa (3.23% 95% CrI 2.21% to 5.07%) (S2 Fig).

The prevalence of glaucoma (unclassified glaucoma) is higher in males (2.94% 95% CrI 2.20% to 4.66%) than in females (2.36% 95% CrI 1.67% to 3.68%) – as reported by six out of the nine studies that reported prevalence of glaucoma.

The model diagnostics are presented in S2 Fig. The graphical representations for the posterior predictive density of $\mu$ and $\tau$ parameters of glaucoma (unclassified glaucoma), POAG, PACG and SG suggest normality of posterior distribution of $\mu$ and a slight positively skewed distribution for $\tau$ (S2 Fig). The assessment of model convergence (with Rhat = 1) for glaucoma (unclassified glaucoma), POAG, PACG and SG, indicate that the algorithm found the optimal solution and that

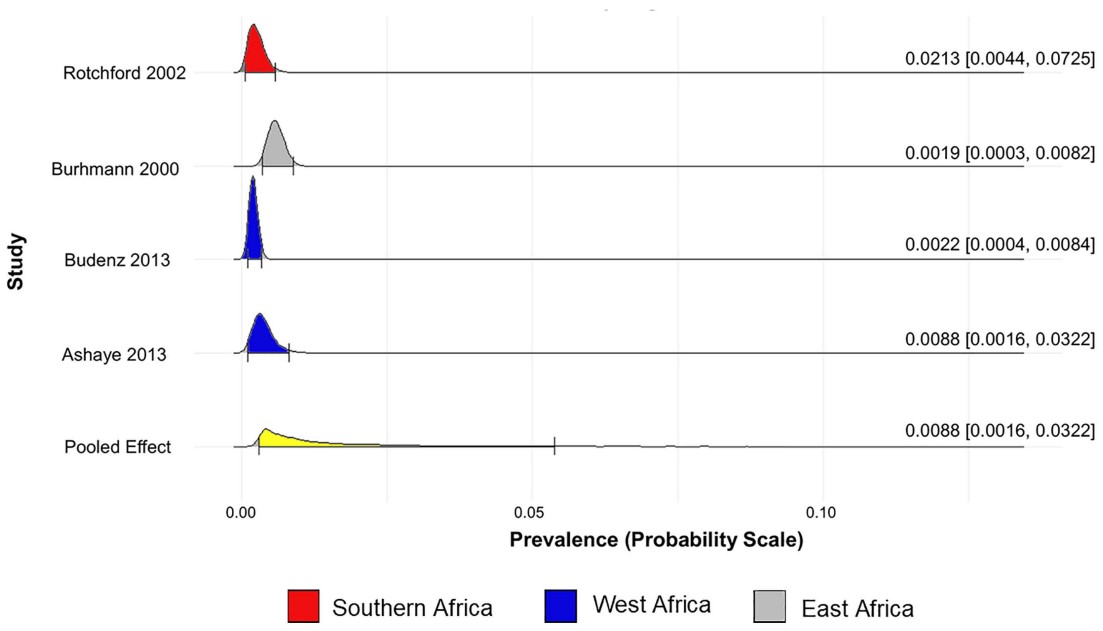

**Fig 5. Forest plot of the prevalence of secondary glaucoma in Africa.**

the models converged, which suggests that the model parameters are trustworthy, as illustrated in the Trace Plot and in the PPcheck plot (S3 Fig).

### Meta-regression

The log-odds of glaucoma (unclassified glaucoma) prevalence in Southern Africa was 0.42 (95% CrI: −0.76, 1.66) higher compared to the reference location (East Africa). This means that, when expressed on a logarithmic scale, the odds of having glaucoma in Southern Africa are estimated to be 0.42 units higher than in East Africa. However, because the credible interval includes zero, this difference is not statistically significant. Similarly, the log-odds of glaucoma prevalence in West Africa was 0.29 (95% CrI: −0.71, 1.22) higher compared to East Africa. Although this effect size is smaller than that for Southern Africa, the interval still includes zero, so there is no statistically significant difference.

The effect of the year of study on glaucoma prevalence was minimal, with a log-odds estimate of −0.01 (95% CrI: −0.05, 0.08). This indicates that there is no meaningful change in glaucoma prevalence over time during the study period.

The log-odds of primary open-angle glaucoma (POAG) prevalence in Southern Africa was 0.08 (95% CrI: −0.65, 0.86) higher compared to the reference location (East Africa), which again is not statistically significant as the interval includes zero. In West Africa, the log-odds of POAG prevalence was 0.82 (95% CrI: −0.11, 1.89) higher compared to East Africa. Even though the effect in West Africa is numerically larger than that in Southern Africa, the credible interval overlaps with zero, meaning this difference is also not statistically significant.

The effect of the year of study on POAG prevalence was negligible, with a log-odds estimate of −0.01 (95% CrI: −0.09, 0.06), suggesting no significant trend over time in POAG prevalence.

### Sensitivity analysis

Switching the parameters for the intercept priors of glaucoma and its subtypes with a more informative alternative (half-normal distribution with a sigma parameter of 1.0) produced minimal variation in the posterior distribution for

glaucoma and POAG, but not for PACG and SG. Similarly, changing to the half-Cauchy prior (sigma parameter of 0.5) did not meaningfully alter the between-study variance posteriors for glaucoma and POAG, giving credence to the stability and robustness of the findings. There were, however, high between-study variances for PACG and SG (Table 2).

## Discussion

This study focused on ascertaining the prevalence of glaucoma in Africa, a continent reported to be disproportionately affected by the disease [47,48]. The study explained the dynamics of the disease prevalence by age, sex, regional block, and country.

Most studies on this subject have almost always focused on the prevalence of the most common subtype of POAG [49]. From the available literature, this is the first time the prevalence of glaucoma including all its major subtypes has been analytically synthesized in Africa.

The meta-analysis estimated the prevalence of POAG, the most prevalent glaucoma subtype in Africa, to be 5.07% based on population-based studies. A high prevalence of POAG has been documented in studies involving black populations in the West Indies and the United States of America. These include the Barbados [50], St. Lucia [51] and Baltimore [52] studies. However, it has been postulated that despite the inclusion of populations of African descent in the Barbados, St. Lucia and Baltimore studies, environmental and genetic disparities may have influenced the observed difference in POAG prevalence between indigenous black Africans and black populations from the West Indies and the United States of America [6]. It is imperative that POAG in Africa receives the requisite attention, given that this category is more insidious and is asymptomatic in nature [53].

There are more males affected by glaucoma than females and this could be attributed to the speculated protective role of estrogen in females, although this postulation is supported by limited evidence [54]. Estrogen may offer neuroprotective effects on the optic nerve by improving blood flow in the inferotemporal retinal artery through its vasodilatory effect [55], while reduced estrogen exposure in men could contribute to their higher glaucoma risk [54]. Male predominance in glaucoma may be partly explained by higher occupational exposure [56]. Although evidence directly linking occupational exposure to male predominance in glaucoma is limited, it has been reported that men are more likely to work in industries involving exposure to physical hazards, such as radiation or air pollutants, which have been associated with increased glaucoma risk [56]. Sex differences in healthcare-seeking behavior may also exacerbate the impact of occupational exposures, as males often delay seeking medical attention, potentially leading to later detection and treatment [57,58]. While these factors may contribute to the observed male predominance, further research is needed to establish definitive links and mechanisms. The higher prevalence of POAG in males underscores the importance of gender-sensitive approaches in glaucoma care. Encouraging men to prioritize routine eye exams could foster early diagnosis and management of glaucoma.

While Southern Africa shows a high prevalence of glaucoma (unclassified glaucoma), West Africa stands out for its particularly high prevalence of POAG. The high prevalence of POAG in West Africa could be the result of an intricate interplay between unique genetic predispositions and environmental stressors [6]. When genetic susceptibilities peculiar to continental Africans encounter environmental insults such as oxidative stress from prolonged exposure to UV light from the sun, the threshold for disease manifestation can be reached more quickly, leading to the characteristic early onset and rapid progression observed in West African populations [59,60]. Glaucoma poses a significant public health challenge in Africa, where the high prevalence rates, especially for POAG, necessitate targeted public health interventions to improve awareness, screening, and treatment access [61]. A considerable number of patients present with advanced disease due to delayed diagnosis, which complicates treatment outcomes [62]. Despite the known prevalence rates, there remains a lack of comprehensive epidemiological data on glaucoma across different African populations. This is particularly evident in the paucity of population-based glaucoma prevalence data for North and Central Africa. Further research is imperative to elucidate the genetic factors contributing to the elevated susceptibility to formulate tailored public health strategies.

The absence of a significant temporal trend in the prevalence of glaucoma (unclassified glaucoma) and POAG indicates that the burden of this disease has remained fairly stable over the analyzed period. A previous Bayesian meta-analysis reported a prevalence of POAG in Africa of 4.20% [30]. This prevalence rate differs from that reported in the present study due to the inclusion of the study by Ashaye et al. [43] in the present study. The stability in temporal trend of glaucoma could largely be attributed to under-resourced healthcare systems, and a general poor knowledge about glaucoma among Africans which hinder early presentation, leading to late detection and timely treatment [63,64]. Low- and middle-income countries face challenges which include limited availability of diagnostic equipment, a shortage of trained eyecare professionals, and financial constraints that substantially restrict access to quality eyecare [65]. While increasing knowledge and awareness is imperative in combatting glaucoma, it should be accompanied by extensive investments in training of primary eyecare practitioners, development of sustainable, widespread screening initiatives, and healthcare infrastructure [65] especially in West Africa, where POAG is most prevalent.

The strengths of the present study include comprehensive search strategies, strict inclusion and exclusion criteria, and robust meta-analytic methods. This study is, however, not without limitations as the limited number of studies could have limited statistical power in sub-group analyses. Furthermore, the limited number of studies precluded sub-group analysis by age-group, which could have offered valuable insights into the distribution of the prevalence of glaucoma and its subtypes by age groups. The inclusion of only nine population-based studies (albeit being the only population-based studies available) over two decades affects the generalizability of findings especially for PACG and secondary glaucoma, where convergence was not reached in some models due to the small number of studies. Although the target acceptance rate was increased from 0.99 to 0.999 and the number of iterations increased from 4,000–10,000, convergence was not attained for the PACG and SG models (Rhat > 1.0). As such, the results of the meta-regression for the PACG and SG subtypes based on the geographic location and year of study covariates were unreliable. Again, while the results of the sensitivity analysis proved robustness and reliability of the glaucoma and POAG estimates from the main meta-analysis, the PACG and SG estimates varied significantly across priors, suggesting unreliability. This could be alluded to the limited number of studies reporting PACG and SG prevalence. Results for PACG and SG should therefore be interpreted with caution. An increase in the number of PACG and SG prevalence studies is also warranted to improve estimate stability and reduce prior dependence. The analysis is deficient in its omission of data from North and Central Africa (population-based prevalence data have not been reported in these regions), thereby restricting the generalizability of the findings. Population-based studies in these regions are needed, as they would provide a more precise estimation of the prevalence of glaucoma in Africa. Relying solely on Google Translate for the translation of non-English articles into English could have introduced potential errors, particularly in the interpretation of complex terms or nuanced descriptions. To mitigate this limitation, a validation step involving cross-referencing the automated translations with professional translations could be conducted to ensure greater accuracy and consistency.

In conclusion, the prevalence rates of glaucoma and POAG are high, with geographical regional variations worthy of note. The stability in the temporal trend of glaucoma may be indicative of challenges in early detection and awareness of the disease. Continued efforts are necessary to implement population-based screening and public health education initiatives to foster early diagnosis and management of glaucoma.

## Supporting information

**S1 Table. Search strategy.**
(DOCX)

**S2 Table. List of studies obtained from literature searches.**
(XLSX)

**S3 Table. Data extracted from the nine primary sources for the Bayesian meta-analysis.**
(XLSX)

**S1 Fig. Risk of bias of included studies.**
(TIF)

**S2 Fig. Marginal effects of glaucoma (unclassified) (A) and primary open-angle glaucoma (B) by geographic location.**
(PDF)

**S3 Fig. Model diagnostics and posterior predictive check plots for glaucoma (unclassified glaucoma), primary open-angle glaucoma, primary angle-closure glaucoma and secondary glaucoma.**
(PDF)

## Acknowledgments

The authors are grateful to Janet Afful, Serwaa Nsiah Appiah, and Lily Kusi for their support.

## Author contributions

**Conceptualization:** Samuel Kyei.

**Data curation:** Randy Asiamah, Gideon Owusu, Patrick Evans Agyiri.

**Formal analysis:** Randy Asiamah.

**Investigation:** Randy Asiamah, Samuel Kyei, Gideon Owusu, Patrick Evans Agyiri.

**Methodology:** Randy Asiamah.

**Project administration:** Samuel Kyei.

**Software:** Randy Asiamah.

**Supervision:** Samuel Kyei.

**Validation:** Randy Asiamah.

**Visualization:** Randy Asiamah.

**Writing – original draft:** Randy Asiamah.

**Writing – review & editing:** Samuel Kyei.

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
