## [Decision Letter · Decision Letter 0]

24 Apr 2025

Dear Dr. Kyei,

Thank you for submitting your manuscript to PLOS ONE. After careful consideration, we feel that it has merit but does not fully meet PLOS ONE’s publication criteria as it currently stands. Therefore, we invite you to submit a revised version of the manuscript that addresses the points raised during the review process.

**ACADEMIC EDITOR: **

Please address reviewers' concerns

We look forward to receiving your revised manuscript.

Kind regards,

Nader Hussien Lotfy Bayoumi, M.D., FRCS (Glasgow)

Academic Editor

PLOS ONE

Journal Requirements:

2. As required by our policy on Data Availability, please ensure your manuscript or supplementary information includes the following:

Reviewers' comments:

Reviewer's Responses to Questions

**Comments to the Author**

1. Is the manuscript technically sound, and do the data support the conclusions?

Reviewer #1: Yes

Reviewer #2: Yes

Reviewer #3: Partly

2. Has the statistical analysis been performed appropriately and rigorously?

Reviewer #1: Yes

Reviewer #2: Yes

Reviewer #3: Yes

3. Have the authors made all data underlying the findings in their manuscript fully available?

Reviewer #1: Yes

Reviewer #2: Yes

Reviewer #3: No

4. Is the manuscript presented in an intelligible fashion and written in standard English?

Reviewer #1: Yes

Reviewer #2: Yes

Reviewer #3: Yes

Reviewer #1: Peer Review Report

This manuscript, "Prevalence of Glaucoma in Africa: A systematic review and Bayesian Meta-Analysis" presents a timely and methodologically rigorous systematic review and meta-analysis on glaucoma prevalence in Africa. The study addresses a critical gap in understanding regional epidemiological trends and provides valuable insights for public health strategies. However, major revisions are needed to enhance clarity and address methodological limitations & contextual interpretation.

Major Comments

1. Acronyms and Definitions

Acronyms (e.g., PACG, SG) are not consistently defined upon first use in the abstract and introduction. Please define all acronyms (e.g., PACG = Primary Angle-Closure Glaucoma; SG = Secondary Glaucoma) when first introduced to improve readability for non-specialist audiences.

2. The introduction is overly verbose, with some paragraphs containing redundant information. Condense lengthy sections (e.g., glaucoma subtypes, risk factors) and ensure a sharper focus on the study’s rationale.

3. Methods

Justification for Bayesian Approach: Briefly justify why Bayesian meta-analysis was chosen over frequentist methods.

Translation Limitations: Acknowledge potential biases/errors from relying on Google Translate for non-English studies. Suggest validation steps (e.g., professional translation for key studies).

Risk of Bias Assessment: Expand on the results of the Joanna Briggs Institute tool. Specify how many studies were classified as high/moderate/low quality and summarize key methodological weaknesses (e.g., sampling bias).

4. Results

Clarity in Presentation: Ensure tables/figures (e.g., Table 1, Figure 2) are explicitly referenced in the text and described in sufficient detail. For example, clarify what "ISGEO" and "VF/IOP" codes represent in Table 1.

Subgroup Analyses: Highlight the limited statistical power for PACG/SG due to small study numbers earlier in the results to contextualize their uncertain estimates.

5. Discussion

Exploration of Sex Differences: While estrogen’s role is discussed, consider alternative explanations for male predominance (e.g., occupational exposures, healthcare-seeking behavior).

Geographical Variations: Expand on potential drivers of higher POAG prevalence in West Africa (e.g., genetic factors, environmental triggers).

Discuss why glaucoma prevalence remains stable despite global advancements in awareness. Link this to systemic barriers (e.g., under-resourced healthcare systems).

6. References: Ensure all in-text citations (e.g., references 42, 43) are included in the reference list. Cross-check numbering to avoid mismatches.

Minor Comments

1. Abstract: Clarify that "unclassified glaucoma" refers to cases not categorized into specific subtypes.

2. Methods: Specify the search period (e.g., "from inception to January 25, 2025" appears in eligibility criteria but is not explicitly tied to database searches).

3. Results: Simplify technical language (e.g., "log-odds estimates") for broader accessibility.

4. Tables/Figures:

• Format Table 1 to improve readability (e.g., align columns, use consistent abbreviations).

• In Table 2, define "Half-Normal(0.5)" & other priors in a footnote for clarity

5. Limitations: Explicitly mention the exclusion of North/Central African studies as a limitation affecting generalizability.

Reviewer #2: please consider the following

first: no abbreviations in the abstract , replace with the full name

second :many difficult words were used in the manuscript might be replaced by simpler ones

third; in the PRISM chart: please replace wrong outcome by different outcome, different study design and different setting

Fourth: add the full title of the figure and its number below the figure

Reviewer #3: Reviewer Comments for Manuscript ID: PONE-D-25-14594

Title: Prevalence of Glaucoma in Africa: A systematic review and Bayesian meta-analysis

General Comments:

This manuscript addresses an important and underexplored topic—the prevalence of glaucoma and its subtypes across African populations—using a systematic review and Bayesian meta-analysis. The methodological approach is sound, and the inclusion of a Bayesian framework is commendable, given the small number of eligible studies and the expected heterogeneity across settings.

However, several methodological clarifications and contextual discussions are needed to enhance the manuscript’s scientific interpretability. I offer the following major and minor suggestions for the authors’ consideration.

Major Comments:

1. Justification of Priors in Bayesian Analysis:

The authors should provide more detailed justification for the choice of prior distributions, particularly the half-normal distribution with a scale of 0.5 for heterogeneity. Although sensitivity analyses were conducted, a rationale for the selection of informative priors and how they align with previous literature or expert consensus would be valuable.

2. Scope and Limitations of Included Studies:

Only nine population-based studies were included over a two-decade span. This small number significantly limits the generalizability of the findings, especially for PACG and secondary glaucoma, where convergence was not reached in some models. This limitation warrants greater emphasis in the discussion.

3. Regional Representation and Gaps:

The analysis lacks data from North and Central Africa, which limits the applicability of the findings to the entire continent. The authors should highlight this gap more explicitly and suggest how future research can address it.

4. Risk of Bias Assessment Interpretation:

The use of the JBI-PCAT tool is appropriate. However, two studies with notable methodological concerns (e.g., recruitment strategy, sample size) were still classified as “low risk.” Further clarification of how the scoring was applied and interpreted would improve transparency.

5. Interpretation of Temporal Trends:

The lack of significant change in glaucoma prevalence over time is an important finding. The authors should explore possible explanations (e.g., stagnant healthcare infrastructure, limited uptake of screening programs) and relate this to policy implications more clearly.

Minor Comments:

1. Language and Grammar:

The manuscript would benefit from careful proofreading to correct minor grammatical issues and improve clarity. For instance, the sentence “The title provided in the protocol was amended to suit available evidence” could be made clearer.

2. Terminology Clarity:

Please define the term “glaucoma (unclassified)” clearly in the methods section and use it consistently throughout the text.

3. Figures and Tables:

Consider incorporating brief descriptions or legends of figures directly in the main text. Table 2 should include an explanation of "CrI" (Credible Interval) for readers less familiar with Bayesian statistics.

4. Discussion on Gender Differences:

The discussion on the higher prevalence of glaucoma among males and the potential protective role of estrogen is intriguing. However, these statements are speculative and should be clearly presented as hypotheses supported by limited evidence.

5. Typographical Errors:

o “Cape Coat” should be corrected to “Cape Coast” in the author affiliation.

o The ISGEO/VF/IOP codes in Table 1 should be briefly explained or a legend added to aid interpretation.

6. Overstatement of Impact (Introduction, Lines 73–76):

The statement that this study “holds great potential in providing the needed information for the realization of Sustainable Development Goal 3… and to reduce untimely deaths resulting from blindness from glaucoma” appears to overstate the manuscript’s impact. While the study’s findings are certainly valuable for public health planning, glaucoma does not directly cause death, and a prevalence estimate alone is unlikely to significantly influence the broader achievement of SDG 3. I recommend rephrasing this sentence to present a more measured and accurate interpretation of the study’s contribution.

7. Undefined Abbreviations in the Abstract:

In the abstract, the abbreviations POAG, PACG, and SG are introduced without being defined. While POAG (Primary Open-Angle Glaucoma) and PACG (Primary Angle-Closure Glaucoma) are commonly used terms in ophthalmology, SG is not widely recognized and may be ambiguous. I assume it refers to Secondary Glaucoma, but this should be clearly stated. Please define all abbreviations at first use in the abstract to ensure clarity for a broader readership.

Recommendation:

With revisions addressing the points above, this manuscript would make a valuable contribution to the literature on glaucoma epidemiology in Africa. I recommend major revision.

**Do you want your identity to be public for this peer review?** For information about this choice, including consent withdrawal, please see our Privacy Policy

Reviewer #1: No

Reviewer #2: **Yes: ** Prof.Dr Heba Elweshahi

Reviewer #3: No

---

## [Author Response · Author response to Decision Letter 1]

3 May 2025

Dear Editor,

RE: Prevalence of Glaucoma in Africa: A systematic review and Bayesian meta-analysis (Submission ID: PONE-D-25-14594).

Thank you for the opportunity to resubmit our manuscript for re-evaluation. Kindly find below a tabulation of point-by-point responses to the review comments for your further action.

Comment Response Page Number

Editor

Comment: Please ensure that your manuscript meets PLOS ONE's style requirements, including those for file naming.

Response: Thank you very much. It has been ensured that our manuscript meets PLOS ONE’s style requirements.

Comment: As required by our policy on Data Availability, please ensure your manuscript or supplementary information includes the following: A numbered table of all studies identified in the literature search, including those that were excluded from the analyses.

Response: Thank you very much. A numbered table of all studies identified in the literature search, including those that were excluded from the analyses, has been provided. For each excluded study, reasons for exclusion have been provided. S2 Table. Pg 9, ln 186 to 187.

Comment: A table of all data extracted from the primary research sources for the systematic review and/or meta-analysis. The table must include the following information for each study:

Response: Thank you very much. A table of all data extracted from the primary sources for the systematic reviews and Bayesian meta-analysis has been provided. Included within the table are the names of the data extractors, the date of extraction, and a confirmation of eligibility for inclusion. S3 Table. Pg 10, ln 198 to 199.

Comment: If data or supporting information were obtained from another source (e.g. correspondence with the author of the original research article), please provide the source of data and dates on which the data/information were obtained by your research group. Response: Thank you very much. This is not applicable to this study as no such situation arose in our case. All data were obtained from the stipulated databases (PubMed, Web of Science and Scopus) and citation searches.

Comment: If applicable for your analysis, a table showing the completed risk of bias and quality/certainty assessments for each study or outcome. Please ensure this is provided for each domain or parameter assessed. For example, if you used the Cochrane risk-of-bias tool for randomized trials, provide answers to each of the signaling questions for each study. If you used GRADE to assess certainty of evidence, provide judgements about each of the quality of evidence factor. This should be provided for each outcome.

Response: Thank you very much. A figure (traffic light plot, S1 Fig) containing a completed risk of bias assessment using the JBI-PCAT tool has been provided as a supporting information and has been reported within the main text. However, certainty of evidence assessment was not assessed, as unlike traditional frequentist meta-analyses, there are no guidelines for assessing certainty of evidence for Bayesian approaches. S1 Fig. Pg 12, ln 204 to 207.

Comment: An explanation of how missing data were handled. This information can be included in the main text, supplementary information, or relevant data repository. Please note that providing these underlying data is a requirement for publication in this journal, and if these data are not provided your manuscript might be rejected.

Response: Thank you very much. An explanation of how missing data were handled has been provided in the Methods section. Pg 5, ln 98 to 99.

Comment: Please include captions for your Supporting Information files at the end of your manuscript, and update any in-text citations to match accordingly.

Response: Thank you very much. Captions for the supporting information have been provided at the end of the manuscript and in-text citations have been updated accordingly. Pg 5, ln 103; Pg 9, ln 187; Pg 10, ln 199; Pg 12, ln 207; Pg 15, ln 227; Pg 15, ln 234; Pg 15, ln 237; Pg 31, ln 556 to 564.

Reviewer 1

Comment: Acronyms and Definitions

Acronyms (e.g., PACG, SG) are not consistently defined upon first use in the abstract and introduction. Please define all acronyms (e.g., PACG = Primary Angle-Closure Glaucoma; SG = Secondary Glaucoma) when first introduced to improve readability for non-specialist audiences.

Response: Thank you very much. Acronyms have now been defined upon first use in the abstract and introduction. Pg 2, ln 35 to 36; Pg 3, ln 50 to 52.

Comment: The introduction is overly verbose, with some paragraphs containing redundant information. Condense lengthy sections (e.g., glaucoma subtypes, risk factors) and ensure a sharper focus on the study’s rationale.

Response: Thank you very much. The introduction has been condensed, and redundant information have been redacted. Also, a sharper focus on the study’s rationale has been ensured. Pg 3, ln 47 to Pg 4, ln 72.

Comment: Justification for Bayesian Approach: Briefly justify why Bayesian meta-analysis was chosen over frequentist methods. Response: Thank you very much. A justification for why Bayesian meta-analysis was chosen over frequentist methods has been provided within the Methods section and has been highlighted in red. Pg 6 ln 129 to Pg 7 ln 139.

Comment: Translation Limitations: Acknowledge potential biases/errors from relying on Google Translate for non-English studies. Suggest validation steps (e.g., professional translation for key studies).

Response: Thank you very much. Potential errors that may arise from relying only on Google translate for non-English studies, as well as a recommended validation step for future studies have been highlighted within the Discussion. Pg 21, ln 350 to 354.

Comment: Risk of Bias Assessment: Expand on the results of the Joanna Briggs Institute tool. Specify how many studies were classified as high/moderate/low quality and summarize key methodological weaknesses (e.g., sampling bias).

Response: Thank you very much. An expansion on the results of the Joanna Briggs Institute tool has been provided within the Results and has been highlighted in red. Pg 12, ln 204 to 207.

Comment: Clarity in Presentation: Ensure tables/figures (e.g., Table 1, Figure 2) are explicitly referenced in the text and described in sufficient detail. For example, clarify what "ISGEO" and "VF/IOP" codes represent in Table 1.

Response: Thank you very much. It has been ensured that tables/figures are explicitly references in the text and described in sufficient detail. Also, a clarification on what "ISGEO" and "VF/IOP" codes represent has been provided in Table 1. Table 1 and 2. Figs 1 to 5

Comment: Subgroup Analyses: Highlight the limited statistical power for PACG/SG due to small study numbers earlier in the results to contextualize their uncertain estimates.

Response: Thank you very much. The limitation of the results of the PACG/SG subgroup analyses in statistical power due to small study sizes has been provided within the Discussion in order to contextualize the result of the analyses instead of the results section, so as not to mix up aspects of the discussion with the results. Pg 20, ln 334 to 337

Comment: Exploration of Sex Differences: While estrogen’s role is discussed, consider alternative explanations for male predominance (e.g., occupational exposures, healthcare-seeking behavior).

Response: Thank you very much for this kind comment. Alternate explanations for male predominance in the prevalence of glaucoma such as occupational exposures and healthcare-seeking behavior have been explored within the discussion and have been highlighted in red. Pg 18, ln 287 to 295

Comment: Geographical Variations: Expand on potential drivers of higher POAG prevalence in West Africa (e.g., genetic factors, environmental triggers).

Response: Thank you very much. An expansion on the potential drivers of higher POAG prevalence in West Africa has been provided within the discussion. Pg 18, ln 300 to Pg 19, ln 306

Comment: Discuss why glaucoma prevalence remains stable despite global advancements in awareness. Link this to systemic barriers (e.g., under-resourced healthcare systems).

Response: Thank you very much. A discussion on why glaucoma prevalence has remained stable, with links to systemic barriers has been provided. Literature available suggest that there exists a generally poor knowledge of glaucoma, and considering the discordance between knowledge and awareness-where knowledge is what informs actions-there is a close link between poor knowledge and late presentation, leading to untimely management. Shear awareness may therefore not translate into reduction in prevalence. Pg 19, ln 320 to Pg 20, ln 328.

Comment: References: Ensure all in-text citations (e.g., references 42, 43) are included in the reference list. Cross-check numbering to avoid mismatches.

Response: Thank you very much for this kind comment. It has been ensured that all in-text citations are included in the reference list, and the numbering mismatch has been addressed. Pg 22, ln 372 to Pg 30, ln 555.

Comment: Abstract: Clarify that "unclassified glaucoma" refers to cases not categorized into specific subtypes.

Response: Thank you very much. It has been clarified within the Abstract and throughout the document that unclassified glaucoma refers to cases not categorized into specific subtypes. Pg 2, ln 34; Pg 19, ln 193; Pg 196, ln 196; Table 2; Pg 15, ln 214; Fig 2; Pg 15, ln 228; Pg 15, ln 232; Pg 15, ln 235

Comment: Methods: Specify the search period (e.g., "from inception to January 25, 2025" appears in eligibility criteria but is not explicitly tied to database searches).

Response: Thank you very much. The search period has now been explicitly tied to database searches. This amendment has been highlighted in red within the Methods section. Pg 16, ln 239; Pg 18, ln 299 to 300; Pg 19, ln 315 to 416; S3 Fig.

Comment: Results: Simplify technical language (e.g., "log-odds estimates") for broader accessibility.

Response: Thank you very much. Technical language have been simplified within the results for better readability.

Comment: Tables/Figures:

• Format Table 1 to improve readability (e.g., align columns, use consistent abbreviations).

• In Table 2, define "Half-Normal(0.5)" & other priors in a footnote for clarity

Response: Thank you very much. Table 1 has been reformatted to improve clarity. Also, the priors have been defined in the footnote to ensure clarity. Table 1; Table 2

Comment: Limitations: Explicitly mention the exclusion of North/Central African studies as a limitation affecting generalizability.

Response: Thank you very much. An explicit mention of the unavailability of population-based studies from North and Central Africa as a limitation affecting generalizability has been made within the discussion and has been highlighted in red. Page 20, ln 346 to Pg 21, ln 350.

Reviewer 2

Comment: no abbreviations in the abstract , replace with the full name

Response: Thank you very much. Abbreviations in the Abstract have now been defined on first mention. Pg 2, ln 35 to 36.

Comment: many difficult words were used in the manuscript might be replaced by simpler ones

Response: Thank you very much. Technical language such as log-odds ratios and credible intervals (as highlighted by Reviewer 1) have been simplified within the manuscript to improve readability.

Comment: in the PRISM chart: please replace wrong outcome by different outcome, different study design and different setting

Response: Thank you very much. Instances of the term “wrong” has been replaced with “different” within the PRISMA chart and Results. Fig 1; Pg 9, ln 183 to 184.

Comment: add the full title of the figure and its number below the figure

Response: Thank you very much for this kind comment. Although this recommendation would ensure accessibility and ease of reading, the journal prohibits it in its guidelines - “Do not include captions as part of the figure files themselves…”. https://journals.plos.org/plosone/s/submission-guidelines. Figure numbers and captions have been incorporated into the main text, following the paragraphs in which they were cited. Pg 6, ln 112; Pg 15, ln 218 to 222.

Reviewer 3

Comment: Justification of Priors in Bayesian Analysis:

The authors should provide more detailed justification for the choice of prior distributions, particularly the half-normal distribution with a scale of 0.5 for heterogeneity. Although sensitivity analyses were conducted, a rationale for the selection of informative priors and how they align with previous literature or expert consensus would be valuable.

Response: Thank you very much. A detailed justification for the choice of prior distributions and how they align with previous literature has been provided within the Methods section and has been highlighted in red. Pg 7, ln 140 to 145.

Comment: Scope and Limitations of Included Studies:

Only nine population-based studies were included over a two-decade span. This small number significantly limits the generalizability of the findings, especially for PACG and secondary glaucoma, where convergence was not reached in some models. This limitation warrants greater emphasis in the discussion.

Response: Thank you very much. The limitation on generalizability of findings, especially for PACG and secondary glaucoma due to the small number of studies has now been emphasized within the discussion and has been highlighted in red. Pg 20, ln 334 to 337.

Comment: Regional Representation and Gaps:

The analysis lacks data from North and Central Africa, which limits the applicability of the findings to the entire continent. The authors should highlight this gap more explicitly and suggest how future research can address it.

Response: Thank you very much. The gap caused by the absence of data from North and Central Africa, as well as a suggestion on how future research can address it has been highlighted within the discussion. Pg 20, ln 346 to Pg 21, ln 350.

Comment: Risk of Bias Assessment Interpretation:

The use of the JBI-PCAT tool is appropriate. However, two studies with notable methodological concerns (e.g., recruitment strategy, sample size) were still classified as “low risk.” Further clarification of how the scoring was applied and interpreted would improve transparency.

Response: Thank you very much. An error was made in reporting the results of the risk of bias assessment and has now been amended. The risk of bias assessment result now reads “All nine studies included in this review were found to be of low risk of bias, with one study (by Ekwerekwu et al.[40]) satisfying eight out of the nine JBI-PCAT items as it failed to describe study subjects and setting in detail. The robust methodological quality across the studies reinforces confidence in the reliability of their findings”. A clarification on how the scoring was applied has been captured within the methods section to improve transparency. Pg 12, ln 204 to 207; Pg 9, ln 177 to 178.

Comment: Interpretation of Temporal Trends:

The lack of significant change in glaucoma prevalence over time is an important finding. The authors should explore possible explanations (e.g., stagnant healthcare infrastructure, limited uptake of screening programs) and relate this to policy implications more clearly.

Response: Thank you very much. A discussion on why glaucoma prevalence has remained stable over time, with links to barriers such as stagnant healthcare infrastructure has been highlighted within the discussion. This has been coupled by relations to policy implications. Pg 19, ln 320 to Pg 20, ln 328.

Comment: Language and Grammar:

The manuscript would benefit from careful proofreading to correct minor grammatical issues and improve clarity. For instance, the sentence “The

---

## [Decision Letter · Decision Letter 1]

14 Jul 2025

Dear Dr. Kyei,

Thank you for submitting your manuscript to PLOS ONE. After careful consideration, we feel that it has merit but does not fully meet PLOS ONE’s publication criteria as it currently stands. Therefore, we invite you to submit a revised version of the manuscript that addresses the points raised during the review process.

**ACADEMIC EDITOR: **

Please attend to the minor comments raised by the reviewer.

We look forward to receiving your revised manuscript.

Kind regards,

Nader Hussien Lotfy Bayoumi, M.D., FRCS (Glasgow)

Academic Editor

PLOS ONE

Journal Requirements:

Reviewers' comments:

Reviewer's Responses to Questions

**Comments to the Author**

Reviewer #1: All comments have been addressed

Reviewer #3: All comments have been addressed

2. Is the manuscript technically sound, and do the data support the conclusions?

Reviewer #1: Yes

Reviewer #3: Yes

3. Has the statistical analysis been performed appropriately and rigorously?

Reviewer #1: Yes

Reviewer #3: Yes

4. Have the authors made all data underlying the findings in their manuscript fully available?

Reviewer #1: Yes

Reviewer #3: Yes

5. Is the manuscript presented in an intelligible fashion and written in standard English?

Reviewer #1: Yes

Reviewer #3: Yes

Reviewer #1: This manuscript makes a valuable contribution to the literature on glaucoma epidemiology in Africa. The authors addressed comments thoroughly & their revisions align well with the peer review recommendations, making the manuscript stronger and more accessible. The responses were systematic & well-documented, demonstrating a commitment to improving the study's quality. While the manuscript is now robust, a few minor tweaks could polish it further:

The term "unclassified glaucoma" is clarified but may be confused with "cases not categorized into specific subtypes." Standardizing this phrasing (e.g., always using "unclassified glaucoma") would avoid confusion.

The forest plots (Figures 2–5) are informative, but adding subtle color differentiation (e.g., for regional subgroups) could enhance interpretability at a glance.

A few sentences remain slightly verbose (e.g., "The necessity for epidemiological, population-based glaucoma studies in these regions is evident…"). Simplifying to "Population-based studies in these regions are needed…" would tighten the prose.

Final Recommendation:

I recommend acceptance pending these small edits & I look forward to seeing it published.

Reviewer #3: (No Response)

**Do you want your identity to be public for this peer review?** For information about this choice, including consent withdrawal, please see our Privacy Policy

Reviewer #1: No

Reviewer #3: No

---

## [Author Response · Author response to Decision Letter 2]

15 Jul 2025

Dear Editor,

RE: Prevalence of Glaucoma in Africa: A systematic review and Bayesian meta-analysis (Submission ID: PONE-D-25-14594R1).

Thank you for the opportunity to resubmit our manuscript for re-evaluation. Kindly find below a tabulation of point-by-point response to the review comment for your further action.

Reviewer 1

Comment: The term "unclassified glaucoma" is clarified but may be confused with "cases not categorized into specific subtypes." Standardizing this phrasing (e.g., always using "unclassified glaucoma") would avoid confusion. Response: Thank you very much. The phrasing (use of “unclassified glaucoma”) has been standardized throughout the manuscript and have been highlighted in red.

Pg 2, ln 34; Pg 9, ln 193; Pg 10, ln 196; Table 2; Pg 15, ln 213, 226, 230, 232; Figure 2; Pg 16, ln 237; Pg 18, ln 297; Pg 313; S2 Fig; S3 Fig

Comment: The forest plots (Figures 2–5) are informative, but adding subtle color differentiation (e.g., for regional subgroups) could enhance interpretability at a glance.

Response: Thank you very much for this kind comment. The subtle color differentiation, signifying regional subgroups, have been added to the forest plots.

Figures 2 to 5

Comment: A few sentences remain slightly verbose (e.g., "The necessity for epidemiological, population-based glaucoma studies in these regions is evident…"). Simplifying to "Population-based studies in these regions are needed…" would tighten the prose.

Response: Thank you very much. The verbose sentence has been rephrased to tighten up the prose.

Pg 20, ln 345 - 346.

Editorial Comments

Comment: If the reviewer comments include a recommendation to cite specific previously published works, please review and evaluate these publications to determine whether they are relevant and should be cited.

There is no requirement to cite these works unless the editor has indicated otherwise.

Response: Thank you very much. The reviewer’s comments do not include recommendations to cite previously published works.

Comment: Please review your reference list to ensure that it is complete and correct.

Response: Thank you very much. The reference list has been reviewed to ensure completeness and accuracy.

Comment: If you have cited papers that have been retracted, please include the rationale for doing so in the manuscript text, or remove these references and replace them with relevant current references.

Response: Thank you very much. No retracted paper has been cited within the manuscript.

Comment: Any changes to the reference list should be mentioned in the rebuttal letter that accompanies your revised manuscript.

Response: Thank you very much. No change has been made to the reference list.

Comment: If you need to cite a retracted article, indicate the article’s retracted status in the References list and also include a citation and full reference for the retraction notice.

Response: Thank you very much. We do not wish to cite a retracted article.

---

## [Editor Report · Decision Letter 2]

4 Aug 2025

Prevalence of Glaucoma in Africa: A systematic review and Bayesian meta-analysis

PONE-D-25-14594R2

Dear Dr. Kyei,

We’re pleased to inform you that your manuscript has been judged scientifically suitable for publication and will be formally accepted for publication once it meets all outstanding technical requirements.

Kind regards,

Nader Hussien Lotfy Bayoumi, M.D., FRCS (Glasgow)

Academic Editor

PLOS ONE
---

## [Editor Report · Acceptance letter]

PONE-D-25-14594R2

PLOS ONE

Dear Dr. Kyei,

I'm pleased to inform you that your manuscript has been deemed suitable for publication in PLOS ONE. Congratulations! Your manuscript is now being handed over to our production team.

Kind regards,

on behalf of

Professor Nader Hussien Lotfy Bayoumi

Academic Editor

PLOS ONE